# The Impact of Nutrition and Intestinal Microbiome on Elderly Depression—A Systematic Review

**DOI:** 10.3390/nu12030710

**Published:** 2020-03-07

**Authors:** Blanka Klimova, Michal Novotny, Martin Valis

**Affiliations:** Department of Neurology of the Medical Faculty of Charles University and University Hospital in Hradec Kralove, Sokolska 581, Hradec Kralove, Czech Republic; novotmi1@fnhk.cz (M.N.); martin.valis@fnhk.cz (M.V.)

**Keywords:** depression, elderly, microbiome, serotonin, tryptophan, gut-brain axis, nutrition, vitamin

## Abstract

The aim of this review is to systematically review the evidence whether proper nutrition has a positive impact on the prevention or decline of depressive symptoms among elderly people. In addition, possible connections between nutrition, microbiome, and serotonin molecules and its tryptophan precursor are discussed. The methodology follows the PRISMA guidelines, including the PRISMA flow chart. The authors systematically reviewed peer-review, English-written articles published in Web of Science and PubMed between 2013 and 2018. The findings of six original articles, detected on the set inclusion and exclusion criteria, indicate that there is an association between nutrition and depressive symptoms in the target group, i.e., that proper nutrition has a positive impact on the prevention or reduction of depressive symptoms among elderly people. The findings also reveal that there is a considerable correlation between the intakes of vitamin B and a decrease in the prevalence of depressive symptoms. Furthermore, sufficient nutrient intake of tryptophan appears to be an important factor in terms of nutrition and serotonin levels in the body. The authors consider it important to explore associations between the overall dietary intake and depression since diets are not consumed as individual nutrients. Returning to preventive approaches seems to be a rational way to promote the mental health of seniors. Future studies thus need to include interdisciplinary collaboration: from a good diagnosis of the disease by a psychiatrist, through an analysis of the need for nutrient metabolism by a biochemist to the development of a nutritional plan by a nutritional therapist. The limitations of this review consist in a relatively small number of the studies on this topic, including just few randomized controlled trials, which are a guarantee of efficacy and objectivity in comparison with cross-sectional studies.

## 1. Introduction

At present, the global population is aging. One of the most serious problems of the aging population is depression. In fact, depression together with dementia is the most common disorder in this age group. It affects between 5% and 7% of the world’s population [1]. Depression can be perceived as a mood disorder that results in a continuous feeling of melancholy, apathy, and indifference. In severe cases, it can result in suicide, especially in elderly white people [2,3]. The suicide rate is two times higher among elderly people at the age of 80-84 years than among other general population. Therefore, the National Institute of Mental Health sees depression in people at the age of 65 and over to be a major public health problem [4]. Depression seems to be one of the key economic issues as far as the healthcare costs are concerned. By 2030, depression is estimated to be the principal reason of disease burden in middle- and higher-income countries. Depression in later life is connected with frailty, a higher death rate, and poorer outcomes from physical illness [5]. 

In case of older people, depressive symptoms are often overlooked and untreated because they coincide with other problems/multiple diseases older people may have [2]. Currently, there is no reliable cure for this disorder, but there are a few strategies that can help in the treatment of its symptoms. These comprise both pharmacological (e.g., antidepressants) and non-pharmacological (e.g., psychotherapy) approaches [6]. One of the critical moments of modern pharmacotherapy is that up to a third of patients with major depressive disorder do not respond adequately to treatment. Currently, there is an intensive effort in the field of science to find fast-acting and potent antidepressants, with ketamine and its derivatives appearing to be the most promising at present [7]. One of the non-pharmacological approaches, as research studies [8,9,10,11] indicate, is nutrition as part of a healthy lifestyle. Research shows that a proper diet and a healthy lifestyle should be seen as a possible aim in the prevention of depression [12]. 

Recently discovered relationships between human intestinal microbiome and the brain seem likely to result in psychiatric disorders, including the mentioned depressive disorder [13]. Research shows that certain types of bacteria or their metabolic products may be associated with depression and reduced quality of life [14]. The intestinal microbiome participates in the proper functioning of the intestine, functions as part of the immune system, and it also has the ability to produce neuroactive molecules. If the intestinal microbiome is disrupted, CNS function disorders may also occur [15].

The fetal digestive tube is sterile. First, the colon is exclusively colonized by microorganisms from the genera Streptococcus or Staphylococcus. These strains subsequently create a suitable environment for other microorganisms and the colon is colonized, for example, by the genera Bacteroides, Bifidobacterium, or Lactobacillus [16]. Due to the huge amounts of bacteria present in the intestine, it was not possible to distinguish between species by commonly used cultivation methods, therefore, only with the advent of molecular genetics methods it was found that only seven bacterial strains (out of 52 thus far defined) colonize the human intestinal tract. The microbiome consists of the following four strains: Firmicutes (e.g., genera of Staphylococcus, Lactobacillus, Streptococcus), Actinobacteria, Bacteroidetes, and Proteobacteria [17,18]. Although each individual has a specific microbiome, there are models that try to classify groups of people according to their microbiome composition into three types, the so-called enterotypes. Enterotype 1 has Bacteroides as its best indicator, enterotype 2 is driven by Prevotella, and enterotype 3 is distinguished by the excess of Firmicutes [19,20]. It should be noted that there are still controversies around this classification. It is believed that the division into different enterotypes is independent of sex, age, BMI, or race, but it is closely related to diet and eating habits: enterotype 1 is predominantly high in protein and animal fat, enterotype 2 is predominant in sugar, and enterotype 3 is predominant in alcohol and polyunsaturated fats [17].

It is well known that the intestine, in order to function properly, receives regulatory signals from the central nervous system (CNS). Over the past two decades, research studies have proved that the intestine is also capable of transmitting signals to the CNS and is received by the brain. In the literature, this term is termed "gut-brain axis". This two-way communication channel includes both the neuronal pathway and the endocrine, nutritional, and immune pathways [21]. Under physiological and homeostasis conditions, the system is balanced in all directions. However, if, for example, there is a microbial overgrowth and pathological situation in the intestine, this delicate balance is disrupted, and this can manifest both at the gastrointestinal or immune level, and at the central level [22]. Thus, an imbalance in the human intestinal microbiota may give rise to the development of a seemingly unrelated central nervous system disease, such as depression.

The purpose of this study is to examine whether proper nutrition has a positive effect on the prevention or reduction of depressive symptoms among elderly people. Moreover, the role of microbiome in the possible development of a depressive disorder is discussed in general (regardless of age) in this article.

## 2. Methods 

The methodology follows the PRISMA guidelines. The authors systematically reviewed peer-reviewed, English-written articles published in Web of Science and PubMed between 2013 and 2018, since several review studies [23,24,25,26] on this topic had been published before. The searched collocations included the following words: nutrition and elderly with depression, nutrition and older people with depression, diet intervention and elderly with depression, dietary deficiencies and elderly with depression, diet intervention and older people with depression, dietary deficiencies and older people with depression, vitamin intake and elderly with depression, microbiome and elderly with depression, microbiome and older people with depression. The terms used were searched using and to combine the keywords listed and using ‘or’ to remove search duplication where possible. In addition, a backward search was also performed, i.e., references of detected studies were evaluated for relevant research studies that authors might have missed during their search. In addition, a Google search was conducted in order to detect unpublished (gray) literature. 

The authors performed an independent quality assessment of these studies. They read the articles to assess eligibility and to determine the quality. The basic quality criteria were adequately described study design, participant characteristics, control conditions, outcome measures, and key findings, with special focus on statistically significant differences (Table 1). The authors selected these basic quality criteria using Health Evidence Quality Assessment Tool for review articles.

The primary outcomes of this review were as follows:to explore the impact of nutrition and intestinal microbiome on elderly depression;to discuss possible associations between nutrition, microbiome, and serotonin molecules and its tryptophan precursor.

## 3. Results

Altogether, 470 titles/abstracts were detected by using the collocation words described above from the database/journal searches. The majority of articles were found in Web of Science (297 studies). In PubMed, 173 studies were found. Another six articles were detected from additional sources, most often from the references of the identified studies. After removing duplicates and titles/abstracts unrelated to the research topic, 142 English-written studies remained. Of these, only 62 articles were relevant for the research topic. These studies were investigated in full and they were considered against the following inclusion and exclusion criteria. The inclusion criteria were as follows: The time span of the issuing the article was limited by January 1, 2013 up to December 31, 2018.Only peer-reviewed English-written full-text journal articles were involved.Only randomized controlled trials and experimental/cross-sectional studies were included.The primary outcome focused on the association of nutrition and depressive symptoms.The subjects had to be at the age 50+ with or without depression.

The exclusion criteria were as follows:The study protocols, e.g., [27], the studies focusing on other aging diseases in which depression was comorbidity, e.g., [28,29], the studies which contain subjects of all ages, e.g., [30,31], the studies which were aimed at dietary counselling as an intervention, e.g., [32], behavioral studies, e.g., [33], the studies with the reverse hypothesis, e.g., [34], and the review studies, e.g., [23,24,25], were also excluded.

Considering the above described criteria, six articles were eventually included into the final analysis. Figure 1 below describes the selection procedure.

The authors altogether identified six original articles on the research topic. Only two studies [35,36] were randomized controlled trials (RCT) and four studies [37,38,39,40] were cross-sectional studies. Three of these studies originated in Asia (Malaysia, Japan, and South Korea) [36,39,40], one in Australia [35], and two in Europe (Croatia, Norway) [37,38]. Two studies [38,39] examined the relationship between the vitamin intake and depressive symptoms among elderly people; one study [35] examined the association between ω-3 fatty acid supplements and their effect on elderly at risk for depression; one study [37] concentrated on the connections between psychological distress and diet patterns; one study [36] explored fasting and calorie restriction dietary regime and its impact on mood in aging people; and one study [39] examined the predicting factors of depression, including nutrition, in elderly people. The subject samples ranged from 31 elderly to 11,621 older people. They were with or without depressive symptoms. The subjects were usually at the age of 65+ years, apart from one study [36]. The assessments were conducted by standardized methods, which comprised the Geriatric Depression Scale, statistical analyses, and questionnaires. 

The main strengths of the detected research studies are their effort for the objective assessment. However, the limitations include differences in methodologies of the selected studies and particularly the fact that in the cross-sectional studies, it is not likely to identify connections between diet patterns/nutrition intakes and depressive symptoms among elderly people. In some studies, there were small sample sizes, insufficient representativeness, or short duration of RCT. 

Nevertheless, the results of all studies indicate that there is an association between the nutrient intake and depressive symptoms among elderly people. This means that a proper nutrient intake may have a positive impact on the prevention/reduction of depressive symptoms in aging population. Table 1 below gives a summary of the key results from the selected articles. The findings are outlined in alphabetical order of their first author.

**Table 1 nutrients-12-00710-t001:** Overview of six selected studies exploring the effect of nutrition on the prevention or reduction of depressive symptoms among elderly people.

Author	Objective	Intervention Period, Type(s) of Nutrition	Number of Subjects	Main Outcome Measures	Findings	Limitations
Duffy et al. [35] RCT (Australia)	To explore the impact of ω-3 FA supplementation on in vivo GSH concentration in elderly people at risk for depression.	12 weeks; four 1000-mg ω-3 fatty acid (FA) supplements daily (intervention g.), placebo (control).	51 subjects, mean age: 72.2 years.	Magnetic resonance spectroscopy, medical, neuropsychological, and self-report assessments, Patient Health Questionnaire.	ω-3 FA supplementation can decrease oxidative stress mechanisms.	Not reported.
Grønning et al. [37] Cross-sectional study (Norway)	To examine the connections between psychological distress and diet patterns.	Irrelevant for this type of study.	11,621 participants at the age of 65+ years.	Hospital Anxiety and Depression Scale, multivariable regression analyses.	Applying a healthy diet is connected with less anxiety and stress in older adults.	Questionnaires are all based on self-report, and the study has no objective measures of food intake or biological data.
Hussin et al. [36] RCT (Malaysia)	To explore dietary effectiveness in enhancing mood states and depression status in older adults.	Three months; reduction of 300–500 kcal/day.	31 healthy males (Mean±SD), aged 59.7±6.3 years.	Profile of Mood States, Beck Depression Inventory-II, Geriatric Depression Scale, statistical analysis.	Fasting and calorie restriction dietary regime is efficient in enhancing mood states and nutritional status among older men (p<0.05).	Small sample size, short duration.
Miskulin et al. [38] Cross-sectional study (Croatia)	To determine the frequency of vitamin B12 deficiency among older adults and to assess whether there is an association between this deficiency and depressive symptoms in these people.	Three months; irrelevant for this type of study.	140 subjects, mean age 71.0±6.7 years.	Questionnaire, competitive immunoassay vitamin B12 kit.	Depressive symptoms appeared in 100.0% (10/10) people with the vitamin B12 deficiency.	Not a randomized controlled trial (no assessment of causality), socio-economic status was not measured.
Nguyen et al. [39] Cross-sectional study (Japan)	To explore the association between vitamin consumption and depressive symptoms in Japanese adults.	Irrelevant for this type of study.	1634 subjects at the age of 65+ years.	Self-administered questionnaires, interviews, comprehensive health examination, Geriatric Depression Scale, statistical analysis.	There are connections between vitamin deficiencies and depressive symptoms in women and overweight older adults.	Not an RCT (no assessment of causality), self-reported assessments might be bias.
Park et al. [40] Cross-sectional study (South Korea)	To examine the dominant and envisaging factors of depression in older Korean people.	Irrelevant for this type of study.	258 subjects at the age of 65+ years.	Questionnaires, Geriatric Depression Scale, statistical analysis.	In men—deficient protein consumption, suffering from more chronic diseases; in women—deficient vitamin B6 consumption, lower cognitive functions, and higher social isolation.	The same as above + limited representativeness of the sample.

## 4. Discussion

The findings of all six studies described in Table 1 indicate that there is an association between nutrition and depressive symptoms in the target group, i.e., that proper nutrition has a positive effect on the prevention or reduction of depressive symptoms among elderly people. In other words, an unhealthy nutrient intake is considered to be a modifiable risk factor for depression among elderly people [36]. The findings also reveal that there is a significant correlation between the intakes of vitamin B and an increase in the prevalence of depressive symptoms [38,39,40]. This has also been confirmed by other research studies [41,42]. 

Furthermore, depressive symptoms are connected with increased oxidative stress [43], which, as the findings of this review showed, could be reduced by ω-3 fatty acid supplements [34]. ω-3 fatty acid is especially essential for the maintenance of cell structure and have anti-inflammatory effects. Duffy et al. [34], in their study, proved that ω-3 fatty acid supplements may prevent the worsening of subclinical depressive symptoms. On the one hand, the protective effect of 3-3 supplementation is described, but on the other hand, it appears that polyunsaturated fatty acids (PUFAs, -3 and -6) may also play an important role in the pathophysiology of depression. Polyunsaturated fatty acids are involved in the structural and functional regulation of neurons, glial cells and endothelial cells in the brain. PUFAs undergo enzymatic metabolism in the body (cyclooxygenase, lipoxygenase, cytochrome P450) to form other bioactive molecules [44]. It is generally accepted that neuro-inflammation plays an important role in the pathophysiology of depression. Recent studies have linked inflammation with polyunsaturated fatty acids, respectively, with their metabolites. Soluble epoxy hydrolases (sEH), which are commonly present in all organisms, metabolize epoxy fatty acids with an anti-inflammatory effect to the corresponding 1, 2-diols with pro-inflammatory properties [45,46]. 

The results of Hussin et al.’s [36] study also show that fasting and calorie restriction dietary regime (with the retention of protein, vitamin, mineral, and water intake) are effective in improving mood states and nutritional status. Hussin et al. [36] also claimed that fasting is associated with increased brain availability of neurotrophic factors such as serotonin, whose deficiencies are often connected with depression [11,47,48].

The findings also point out gender differences in food preferences [37,49]. Older females seem to have a healthier diet pattern than older males, and therefore, they are less psychologically distressed. Nevertheless, interestingly enough, depression is more common in females than males, which might be associated with stressful life events and higher sensitivity to seasonal changes [50]. 

In addition, the link between nutrition, human intestinal microbiome, enteric nervous system, central nervous system, and depression could be significant, signaling the molecule serotonin or its tryptophan precursor, regardless of age. Tryptophan is known to have a direct effect on the dietary composition of microflora [51,52]. Tryptophan cannot be created by the human body itself; its presence is fully dependent on its dietary intake. One of the functions of tryptophan in the human body is that it serves as a precursor for the synthesis of the neurotransmitter serotonin. Interestingly, the conversion of tryptophan to serotonin occurs predominantly in the intestine, in the enterochromaffin cells of the intestinal mucosa. Therefore, the vast majority of serotonin is not found in the CNS but in the gastrointestinal tract [53]. The serotonin precursor tryptophan is converted by the enzymes to serotonin.

Serotonin is one of the neurotransmitters and, in addition to its well-known effects in the CNS, it has effects that are also implicated in the gastrointestinal tract (GIT) due to the present serotonin receptors [54]. In GIT, it is mainly used to control secretion, vasodilation, peristalsis, pain perception, and nausea. Serotonin levels vary depending on plasma tryptophan levels and on the current availability of tryptophan contained in nutrition [55]. In addition, the gut-brain axis plays an important role, as recent research revealed. The findings show that the metabolism of tryptophan and serotonin in mice have been able to be influenced by administering Lactobacillus reuteri. Even in the animal model, the authors talk about the antidepressant effect [56].

There is another serotonin-independent relationship between tryptophan and affective disorders. The second metabolic pathway of tryptophan in the human body is towards the cynurenine product. It is then metabolized by two different pathways: either leading to kynurenic acid or quinolinic acid [57]. Thus, in susceptible populations, a decrease in tryptophan plasma concentration can cause changes in moods such as depression [52].

Depression can be seen as a combination of disrupted regulations in the neuroendocrine, immune, metabolic, and neurotransmitter systems. Recent preclinical studies suggest that at least some of these processes may be modulated by the intestinal microbiota, or by its changes [58]. In addition to the generally accepted neurotransmitter serotonin, anxiety and depression may be associated with a disorder of another neurotransmitter GABA (gamma-aminobutyric acid) system. Strains of bacteria, such as Lactobacillus and Bifidobacteria, that can synthesize GABA from glutamate (an amino acid from food) are described. Lactobacillus rhamnosus has been shown to reduce anxiety and depressive behavior, while increasing GABA levels in the hippocampus [16,58].

Depression episodes may be also related to hypothalamic-pituitary-adrenal axis hyperactivity. It has been shown that the intestinal microbiota plays an important role in programming this axis in the early stages of life, but also generally in stress reactivity throughout life. The presence of intestinal microbiota at an early stage of life is very important for proper brain development, but the absence of intestinal bacteria during development may negatively affect the hypothalamic–pituitary–adrenal (HPA) axis [59].

Furthermore, antibiotics are widely used worldwide. They are effective in indicated cases but can also be potentially harmful drugs. There are two recent studies of antibiotic administration to animal models and gut-brain monitoring, in which the effect of intestinal microbiome in anhedonia-like phenotypes in mice after chronic social defeat stress (CSDS) [60] was observed on stress resilience using a rat learned helplessness paradigm [61]. The authors made very interesting findings, in that the antibiotic-induced microbiome depletion contributed to the resilience to anhedonia in mice subjected to CSDS, and that abnormal composition of gut microbiota contributes to susceptibility versus resilience to learned helplessness in rats.

Another factor that may contribute to the development of depression is an increase in intestinal permeability with subsequent bacterial translocation. In depressed patients, inflammatory manifestations in the gastrointestinal tract may increase. Typically, elevated levels of inflammatory biomarkers, specifically IL-6, TNF-α, or C-reactive protein, can be detected [16,62,63]. The theory of the impaired intestinal barrier and depression is also supported by Maes et al. [64], who induced intestinal mucosal dysfunction through stress, thereby increasing translocation of gram-negative bacteria, which induced greater HPA axis activity, thereby promoting depression [16].

Immunological reactions also play an important role in affective disorders. Significantly higher serum concentrations of IgM and IgA (against lipopolysaccharide walls of Gram-negative bacteria walls) were found in depressed patients compared to healthy controls [16,58,63]. It is likely that increased translocation of gram-negative bacteria enhances the immune response.

Overall, it seems important to make people aware of the value of healthy nutrition in the prevention of depressive symptoms since it is not only beneficial for their health-related quality of life, but it is also cost-effective [65]. It can be effectively done through public health education or individual consultation with nutritional specialist as research illustrates [66,67]. The studies also show that combined multi-domain interventions (e.g., a healthy diet, physical exercises, and cognitive training) produce a bigger effect than single domain interventions [67,68]. In addition, the most recent project MooDFOOD [9] aims to provide an insight into the causality of the link between diet and depression, and underlying pathways on the basis of both short-term and long-term research studies and transform the acquired knowledge into nutritional strategies, which could serve as guidelines and practical tools to guide policy at EU and Member State levels.

As with physical disorders, mental disorders are affected by nutrition. In the context of an increasingly aging population, the cost of public health budgets is rising. Returning to preventive approaches seems to be a rational way to promote the mental health of seniors. Given that human nutrition is very easy to adjust by simply changing the diet (as compared to other risk factors for depression), one of the first steps should include implementing a sustainable and tolerable nutrition plan by an appropriate expert. It is well known that directive-based diets do not work in the long run and people are unable to sustain them for their lives. The plan should be the result of close cooperation between the patient and the nutritional specialist focusing on the geriatric population. This is also because the evaluation of eating habits is very demanding in the elderly—these patients often have lower cognitive abilities, are burdened with polypragmasia, and are polymorbid. 

Currently, professional literature is struggling with the lack of quality studies on this topic, especially in the long run. Many studies lack robust diagnosis of mental illness, and in others, eating habits have not been carefully evaluated. Longitudinal studies will be needed to explore these complex relationships. Future studies will need to include interdisciplinary collaboration: from a good diagnosis of the disease by a psychiatrist, through an analysis of the need for nutrient metabolism by a biochemist, to the development of a nutritional plan by a nutritional therapist. The overall supervision of the patient in terms of polymorbitis and polypragmasia should be covered by a geriatrician. With adequate funding for such projects, the association between diet and depression at a late age can be explained and preventive approaches may reduce the impact on the public budget.

The limitations of this review include a relatively small number of the studies on this topic, which might also be caused by selecting the studies within a certain time period. In addition, there were just few RCT, which are a guarantee of efficacy and objectivity in comparison with cross-sectional studies. 

## 5. Conclusions

The findings of this review reveal that there is an association between the nutrient intake and depressive symptoms among elderly people. This means that a proper nutrient intake (e.g., vitamins or minerals) may have a positive effect on the prevention/ reduction of depressive symptoms in the aging population. However, it is important to study associations between the overall dietary intake and depression since diets are not consumed as individual nutrients.

Dysfunctional intestinal microbiome may be associated with behavioral disorders such as depression. This connection stems from a recently characterized two-way communication channel between the intestine and the brain, mediated by neuroimmune, neuroendocrine, and sensory neural pathways. Serotonin or its precursor tryptophan seems to be an important molecule.

Furthermore, in the geriatric population, depression and malnutrition go hand in hand and seem to be an interrelated relationship. Depression leads to changes in appetite; appetite is often reduced and the older population suffers from malnutrition, which only intensifies psychiatric symptomatology (depression, irritability, and stress). Dietary supplements containing, for example, vitamin D, folate, magnesium, zinc, and unsaturated fatty acids, do not appear to improve depression in the elderly. It seems much more likely that a comprehensive assessment of a nutritional plan and setting up a suitable diet already at a younger middle age could be the right preventative pathway. This also applies to the case of malnourished elderly patients, where an intensified diet plan drawn up by a specialist and supervised, for example, by a carer, is needed. There is still a consistent recommendation that the Mediterranean diet and regular physical activity together with non-smoking are associated with reduced depressive symptoms later in life, while some dietary supplements (consult above) can positively complement a healthy lifestyle.

Future research should include more randomized controlled studies, which would conclusively prove the efficacy of nutrition in the prevention of depressive symptoms.

## Figures and Tables

**Figure 1 nutrients-12-00710-f001:**
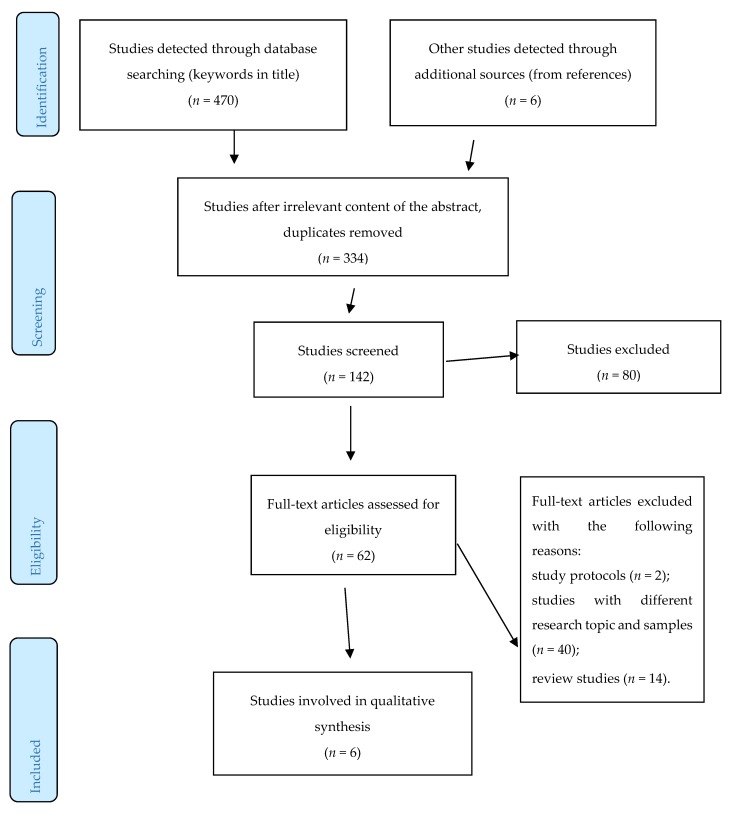
An overview of the selection procedure.

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
