# Peer review of "The Impact of Nutrition and Intestinal Microbiome on Elderly Depression—A Systematic Review"

_nutrients, 2020, doi:10.3390/nu12030710_

Round 1

Reviewer 1 Report

1) The manuscript need proofreading by proficient English speaker. This is already evident since the title, which I would rather rephase as follows: “The impact of nutrition and intestinal microbiome on elderly depression. A systematic review”.
2) The abstract should be a structured one. The rationale and background should be just one or two sentences. The methods should mention the PRISMA guidelines. The results should be as much quantitative as possible, even if this is a systematic review instead of meta-analysis. The limitations are not presented. The discussion should briefly mention the impact for the clinical practice and forthcoming studies on the matter.
3) Methods: there is a coarse mistake: the number of the screened studies or included papers should be part of the results! The authors are strongly encouraged to follow the PRISMA checklist in a very rigorous manner… Please use bullet points for the primary outcomes you planned to extract into advance. The methods are totally flawed. Again, please take a look at good PRISMA-complaint reports already appraised in the literature, even on other topics, to set it-up correctly.
4) There are also very gross mistakes in the flow chart: for example, “they” should read “the”, but no number is given in the box. The excluded studies should be more concisely summarized.
5) The conclusions are too concise and poorly focused with respect to the clinical implications for treatment and prevention.

Reviewer 2 Report

The authors reviewed the role of nutrition in elderly depression. This review is well written, and is of interest for the readers of the journal. The following concerns should be addressed .

Minor concerns:

1) Omega-3 polyunsaturated fatty acids (PUFAs) play a role in elderly depression. Some preclinical findings suggest the role of PUFA metabolism in depression. The following articles should be cited and discussed.

  a) Ren Q, et al. PNAS USA 2016.

  b) Hashimoto K. Front Pharmacol. 2019

  c) Atone J, et al. Prostagrandins Other Lipid Mediat. 2019.

2) Preclinical data: Stress plays a role in the depression. The following articles on the role of gut microbiota in depression-like phenotype should be cited and discussed.

 a) Wang S, et al. J Affect Disord. 2020.

 b) Zhang K, et al. Transl Psychiatry 2019.

 c) Yang C, et al. Transl Psychiatry 2019.

 d) Xie R, et al. J Psychiatr Res 2020.

Round 2

Reviewer 1 Report

Thank you for incorporating my suggested editing.